# Copper-To-Zinc Ratio as an Inflammatory Marker in Patients with Sickle Cell Disease

**Mathias Abiodun Emokpae \***  **and Emmanuel Bamidele Fatimehin**

Department of Medical Laboratory Science, School of Basic Medical Sciences, College of Medical Sciences, University of Benin, Benin City, Nigeria; biodunemokpae@yahoo.com
\* Correspondence: mathias.emokpae@uniben.edu

**Abstract:** Sickle cell disease (SCD) is an inherited disorder and a major health challenge in Nigeria. Micronutrient deficiencies often associated with the disorder may cause inflammation and abnormal metabolism in the body. The copper-to-zinc ratio is a more relevant diagnostic measure than the concentration of either metal alone in clinical practice. This study seeks to evaluate serum levels of c-reactive protein (CRP), copper, zinc and the copper-to-zinc ratio, and to correlate the latter with CRP in adult subjects with SCD. Serum copper, zinc, CRP and plasma fibrinogen were assayed in 100 confirmed SCD patients in steady clinical state and 100 age- and sex-matched subjects with normal hemoglobin. Serum copper and zinc were assayed by the colorimetric method using reagents supplied by Centronic, Germany, while CRP and fibrinogen were assayed using reagents supplied by Sigma (St. Louis, MO, USA) and Anogen (Ontario, Canada), respectively. The copper-to-zinc ratio was calculated from serum levels of copper and zinc. The measured parameters were compared between the groups using the Students *t*-test, and the Pearson correlation coefficient was used to relate CRP with the other parameters. Serum copper, CRP, fibrinogen and the copper-to-zinc ratio were significantly higher ($p < 0.001$), while zinc level was lower in SCD patients than in controls. Serum CRP concentration correlated with copper (r = 0.10; $p < 0.02$), zinc (r = −0.199; $p < 0.05$) and the copper-to-zinc ratio (r = 0.312; $p < 0.002$), but the correlation between CRP and fibrinogen was not significant. Inflammation may modulate copper and zinc homeostasis, and the copper-to-zinc ratio may be used as a marker of nutritional deficiency and inflammation in SCD patients.

**Keywords:** copper; c-reactive protein; inflammation; sickle cell disease; zinc

## 1. Introduction

Sickle cell disease (SCD) is an inherited disorder and a major health challenge in sub-Saharan African, including Nigeria. The condition is characterized by hemolytic anemia and periodic painful crises as a result of the occlusion of small blood vessels due to spontaneous intravascular red blood cell polymerization at reduced oxygen tension [1]. The associated complications of SCD include growth retardation, impaired immune function, acute chest syndrome, abdominal pain [2], proteinuria [3], increased oxidative stress and damage to cell membranes [4]. Micronutrient deficiencies have been reported in patients with SCD [5], a situation made worse by proteinuria [6]. Some authors have suggested that the dietary habits of subjects with SCD met or even exceeded the recommended dietary allowance (RDA) and were, therefore, not different from the general population [7]. The observed deficiencies may be due to abnormal metabolisms of key trace elements in the body which are very important for the maintenance of red blood cell membrane and the growth and development of the body [7].

The copper-to-zinc ratio is a more relevant diagnostic measure than the concentration of either metal alone in clinical practice. Zinc is the second most abundant transition element in the human

body after iron, and it is the only metal that occurs in all enzyme classes. Zinc is involved in several forms of cellular metabolism and plays a role in immune function, wound healing, protein DNA synthesis as well as cell division. It also plays a role in the maintenance of proper sense of taste and smell, and supports growth and development. Zinc possesses antioxidant and antimicrobial properties and confers protection against accelerated ageing [8].

Copper is the third most abundant trace element in the body after iron and zinc. Even though copper is an important micronutrient, only small amounts are needed by the body. It is essential for maintaining the strength of the skin, blood vessels and epithelial and connective tissues. Copper plays a role in the production of hemoglobin, myelin and melanin, and in the proper functioning of the thyroid gland. It also acts as both an anti- and a pro-oxidant [9].

Immune activation, damaged endothelial cells and activation of adhesion molecules often lead to inflammation and the secretion of c-reactive protein (CRP) and other inflammatory mediators in the body. C-reactive protein is a known regulator of inflammation and is the most commonly assayed biomarker of acute and chronic inflammation in clinical practice. It is associated with SCD, and some authors have correlated CRP with other markers of inflammation in SCD patients [10,11].

In healthy individuals, the body has the capacity to manage and control the amount of essential trace elements circulating in the blood and stored in tissue. Dietary essential metals are incorporated into blood if their levels are low, transported into the cells when their cellular levels are depleted, or eliminated when blood and cellular levels are adequate or in excess [12]. When this system fails to function optimally, abnormal levels and imbalances in their ratios occur. This study seeks to evaluate the levels of serum CRP, copper, zinc and the copper-to-zinc ratio. It also correlates CRP with the copper-to-zinc ratio in adult SCD patients.

## 2. Patients and Methods

The study participants were 100 confirmed SCD patients with mean age 18.8 ± 0.9 years in steady clinical state (55 males and 45 females) and 100 apparently healthy subjects with mean age 19.2 ± 0.9 years.

### 2.1. Ethical Consideration

The study protocol was reviewed and approved by the ethics committee of Edo State Hospital Management Board, Benin City (SCC34/2/45 dated 9 September 2014), and all participants gave informed consent before blood samples were collected.

### 2.2. Sample Collection and Preparation

Five milliliters of venous blood was obtained aseptically, of which 2 mL was dispensed into a tube containing 3.8% sodium citrate for the fibrinogen assay and 3 mL was dispensed into a plain container and allowed to clot at room temperature. Both were centrifuged at 3000 rpm for 10 min using a Compact II centrifuge (Pittsburg, PA, USA). The plasma and serum were stored at −20 °C until further use.

### 2.3. Analytical Methods

Serum copper and zinc were assayed by the colorimetric method using kits supplied by Centronic, Germany, while serum CRP and plasma fibrinogen were assayed using reagents supplied by Sigma (St. Louis, MO, USA) and Anogen (Mississauga, ON, Canada), respectively. Commercially available control sera were included in the assay to ensure the accuracy of the analyses.

### 2.4. Statistical Analysis

The results obtained were presented as mean ± standard error of the mean (SEM), and were analyzed using a statistical software package (SPSS version 20, IBM, Chicago, IL, USA). The Student's

*t*-test was used to compare the levels of copper, zinc and CRP in SCD patients and controls. Correlation analysis was carried out using GraphPad Prism 6 (Cal, USA) to test the relationship between the copper-to-zinc ratio and the measured inflammatory marker (CRP).

## 3. Results

The results obtained in this study are presented in Tables 1 and 2. Serum copper, the copper-to-zinc ratio, CRP and plasma fibrinogen were significantly higher ($p < 0.001$) in SCD patients than in controls. On the other hand, serum zinc was significantly lower ($p < 0.001$) in SCD patients than in controls. Table 2 indicates that the copper-to-zinc ratio (r = 0.312; $p < 0.002$) and copper (r = 0.210; $p < 0.02$) correlated positively with CRP, while zinc (r = −0.199; $p < 0.05$) correlated negatively with CRP in SCD patients in steady clinical state. No significant positive correlation (r = 0.048; $p = 0.08$) was observed between plasma fibrinogen and CRP.

**Table 1.** Comparison of measured variables in sickle cell disease patients and controls.

| Parameters | Sickle Cell Disease Patients (*n* = 100) | Controls (*n* = 100) | *p*-Value |
|---|---|---|---|
| Gender | | | |
|     Number of males | 55 | 58 | |
|     Number of females | 45 | 42 | |
| Age (years) | 18.8 ± 0.9 | 19.2 ± 0.9 | 0.80 |
| Serum copper (µmol/L) | 28.92 ± 0.55 | 16.8 ± 0.5 | 0.001 |
| Serum Zinc (µmol/L) | 9.06 ± 0.38 | 13.54 ± 0.22 | 0.001 |
| Copper-to-zinc ratio | 3.16 ± 0.1 | 1.23 ± 0.09 | 0.001 |
| Serum CRP (µg/mL) | 1.14 ± 0.02 | 0.83 ± 0.82 | 0.001 |
| Plasma fibrinogen (mg/dL) | 295 ± 14.8 | 290 ± 16.1 | 0.001 |

**Table 2.** Correlation of copper-to-zinc ratio with C-reactive proteins in SCD patients.

| Correlation | R-Value | *p*-Value |
|---|---|---|
| Copper/c-reactive protein | 0.210 | 0.02 |
| Zinc/c-reactive protein | −0.199 | 0.05 |
| Copper-to-zinc ratio/c-reactive protein | 0.312 | 0.002 |
| Copper-to-zinc ratio/fibrinogen | 0.048 | 0.08 |

## 4. Discussion

Some authors have reported that elevated serum copper levels in the presence of low zinc (high copper-to-zinc ratio) may be a contributing factor to several diseases including hypertension, schizophrenia, autism, fatigue, headaches, childhood hyperactivity, depression, insomnia, senility, premenstrual syndrome and muscle and joint pain [12]. In this study, we observed significantly higher ($p < 0.001$) levels of copper-to-zinc ratio, serum copper, zinc, CRP and plasma fibrinogen in SCD patients than in control subjects with normal hemoglobin. The finding also indicates that the copper-to-zinc ratio correlated positively with CRP. Our finding is consistent with that of a previous study [13]. Some authors reported a significant association between inflammation and the copper-to-zinc ratio in children, while others observed no association [14]. Others reported a significant association between inflammation and serum copper levels in hospitalized patients and adult volunteers. It was suggested that inflammation may influence micronutrient concentrations at CRP levels as low as 0.6 mg/L. Experimental and clinical studies have demonstrated that inflammatory cytokines disturb trace element homeostasis, leading to elevated copper-to-zinc ratios in some disease conditions [15]. Wisniewska et al. [16] reported elevated copper-to-zinc ratios in neonates with early

congenital infections, and also observed that deficiencies of these micronutrients impaired the immune defense and made patients more susceptible to infectious diseases [17–19]. These disturbances in micronutrients in response to inflammation may be contributing factors to a vicious cycle of impaired immune defense and higher risk of infection in susceptible individuals [16]. In a study which evaluated the copper-to-zinc ratio and nutritional status in colorectal cancer patients during the perioperative period, it was observed that elevated copper-to-zinc ratio may be due to a systemic inflammatory response to cancer [20]. An elevated copper-to-zinc ratio was considered as a marker for asthma severity [21], renal disease among patients with peritoneal dialysis [22] and prostate cancer [23]. In the elderly, a high copper-to-zinc ratio was used as a predictive marker for mortality, and was associated with elevated serum CRP and erythrocyte sedimentation rate [24]. Elevated levels of inflammatory biomarkers have been associated with raised levels of plasma copper. Over 90% of circulating copper is bound to caeruloplasmin, and the concentration of caeruloplasmin is elevated in response to inflammation, infection and proinflammatory cytokines. Caeruloplasmin also stimulates the production of acute phase proteins in the liver. Elevated copper-to-zinc ratio has been used as a diagnostic and prognostic marker of inflammation in lymphoma and leukemia, as well as gastric and breast cancer. A higher copper-to-zinc ratio was significantly correlated with undernutrition, oxidative stress, inflammation and depressed immune function [20,22]. An elevated copper-to-zinc ratio also correlated with elevated levels of prostate-specific antigens in subjects with prostate cancer [23], and lipid peroxidation in patients with asthma [21].

The observed elevated levels of CRP in SCD patients were consistent with the results of a previous study [25]. The authors suggested that this may have been due to repeated occlusion of blood vessels and eventual reperfusion of necrotized tissue, resulting in the increased generation of reactive oxygen species. The biological role of CRP in inflammation is not completely understood, but it may stimulate leukocyte migration [26], react with bacterial surfaces to induce phagocytosis [27], enhance immune response [28] and facilitate lymphocyte blast transformation [25]. Other authors have suggested that elevated CRP may have a protective effect on the endothelium due to nitric oxide bioavailability [7]. The high levels of CRP in SCD patients may enable their immune systems to respond better to the numerous challenges associated with SCD [25].

The low levels of zinc observed in SCD patients were consistent with the results of previous studies [1,29]. Asanga et al. [1] reported that 68.3% of SCD patients had hypozincemia; this was attributed to several factors, such as chronic hemolysis, leading to loss of zinc from red blood cells which are vital storage sites for zinc, low dietary intake, increased urinary loss, kidney impairment, disturbance of zinc metabolism and high excretion in sweat [1]. The biochemical sequalae of low serum zinc in SCD patients may be reflected in the low activity of some zinc-dependent-enzymes such as carbonic anhydrase in red blood cells, alkaline phosphatase in neutrophils and thymidine kinase. The effect of low zinc levels was also reported to affect urea cycle enzymes. Decreased activity of ornithine carbamoyl transferase, and increased activity of glutamate dehydrogenase and carbamoyl–phosphate synthase 1, were reported in the liver of zinc deficient rats [29].

The observed high levels of serum copper were, however, inconsistent with data reported by other authors [1], who observed no significant change in the level of copper in SCD patients compared with controls.

## 5. Conclusions

The copper-to-zinc ratio correlated positively with c-reactive protein in patients with SCD. The levels of copper and the copper-to-zinc ratio were significantly higher, while zinc was lower, in SCD patients than in control subjects. Inflammation may modulate copper and zinc homeostasis in SCD patients. It is suggested that the copper-to-zinc ratio may be used as marker of nutritional deficiency and inflammation in patients with SCD.

**Author Contributions:** M.A.E. conceived, designed the experiments, performed the analysis and wrote the manuscript; E.B.F. performed the data gathering, analysis and assisted in the writing of the manuscript. All authors have read and agreed to the published version of the manuscript.

**Funding:** This research received no external funding.

**Acknowledgments:** We appreciate the contributions of the clinical, nursing and Laboratory staff of Central Hospitals, Benin City for their assistance toward the completion of the study.

**Conflicts of Interest:** The authors declare no conflict of interest.

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
