# Peer review of "Copper-To-Zinc Ratio as an Inflammatory Marker in Patients with Sickle Cell Disease"

_sci, doi:10.3390/sci2040089_

Round 1

Reviewer 1 Report

This finding was really correlated or similar to the discovery by the Antwi-Boasiako research group published in the Medicina 2019, 55(5), 180 with the title of Serum Iron Levels and Copper-to-Zinc Ratio in Sickle Cell Disease. This study came with better research designed and more insightful results and conclusion. It also demonstrated that the copper-to-zinc ratio was significantly higher in the SCD patients, suggesting elevated copper-to-zinc ratio may be a biomarker of sickle cell oxidative stress and associated complications. The ratio may also be informative for the management of sickle cell oxidative burden. The authors suggested increased correlates with an Inflammatory Marker in Patients with Sickle Cell. This published article does impair the value of your studies and manuscript.

Author Response

This finding was really correlated or similar to the discovery by the Antwi-Boasiako research group published in the Medicina 2019, 55(5), 180 with the title of Serum Iron Levels and Copper-to-Zinc Ratio in Sickle Cell Disease. This study came with better research designed and more insightful results and conclusion. It also demonstrated that the copper-to-zinc ratio was significantly higher in the SCD patients, suggesting elevated copper-to-zinc ratio may be a biomarker of sickle cell oxidative stress and associated complications. The ratio may also be informative for the management of sickle cell oxidative burden. The authors suggested increased correlates with an Inflammatory Marker in Patients with Sickle Cell. This published article does impair the value of your studies and manuscript. We quite appreciate the comment of this reviewer and it is an opinion. This study evaluated serum levels of c-reactive proteins, copper, zinc and calculated the cu/zn ratio in SCD subjects on clinical steady state and did not include those with vaso-occlusive crisis. The study referred to evaluated serum iron, copper, zinc and calculated cu/zn ratio in SCD patients on steady clinical state and vaso-occlusive crisis. The findings from both studies are similar except for iron and CRP which were not similar to the studies. Whereas we correlated cu/zn ratio with CRP the other correlated cu/zn ratio with iron. Please we would like to know the fundamental flaws detected by the reviewer please. Thank you for the good job.

Reviewer 2 Report

Authors report correlation of serum levels of copper, zinc, copper-to-zinc ratio with  an inflammatory marker (c-reactive protein) in the patients with sickle cell anemia in the population of Nigeria. It is a very interesting study which confirms the correlation of abovementioned parameters towards the inflammation in the patients of sickle cell disease. A similar study has been reported by the Antwi-Boasiako research group published in the Medicina 2019, 55(5), 180 which also includes the serum levels of iron as an additional contributing factor. Moreover, the study reported by the Antwi-Boasiako et al was carried out in Ghana so it will be interesting to correlate both studies to provide a broader overview of the scope of the study. Over all it a very nice manuscript but can be improved.

Author Response

Authors report correlation of serum levels of copper, zinc, copper-to-zinc ratio with an inflammatory marker (c-reactive protein) in the patients with sickle cell anemia in the population of Nigeria. It is a very interesting study which confirms the correlation of abovementioned parameters towards the inflammation in the patients of sickle cell disease. A similar study has been reported by the Antwi-Boasiako research group published in the Medicina 2019, 55(5), 180 which also includes the serum levels of iron as an additional contributing factor. Moreover, the study reported by the Antwi-Boasiako et al was carried out in Ghana so it will be interesting to correlate both studies to provide a broader overview of the scope of the study. Over all it a very nice manuscript but can be improved. We thank this reviewer for job done, but has not suggested specific change(s) to the article. In this study the subjects were all on steady clinical state and none with vaso-occlusive crisis. In addition serum iron was not evaluated in present study. Regards, Emokpae

Round 2

Reviewer 2 Report

The manuscript presents a very nice and extensive study about the correlation od copper to zinc ratio with inflammatory marker in the patients with sickle cell disease. The results are compared with normal subjects and a significant difference is observed. Scientifically it is very nice manuscript. it would be interesting to show a scheme how the factors mentioned in the manuscript interfere with the disease. 

The language of the manuscript can be improved. here are a few suggestions 

Page 1, abstract line 11 and introduction line 41: "ratio of copper to zinc ratio" one of the "ratio" can be deleted

Page 1, abstract line 6: "serum copper and zinc were assayed" it would be better to mention "levels or concentrations" in the sentence.

Page 1, abstract. line 17 1nd 18. I think it is not necessary to mention the the names of the suppliers in the abstract.

 Page 1, line 35,36: "micronutrients deficiency" can be replaced with "micronutrient's deficiency"

Page 2,  line 60 and 62: Please specify what "their " is referring to

Page 2, line 66: spacing in between 18.8±0.9 and years

Page 2, line 74: inconsistent writing style "five mL"  and "2mL". please be consistent with the way of presentation.

Page 2, line 75: How much quantity of 3.8% sodium citrate was used? please mention

Page  2, line 76: "both were centrifuges at " can be replaced with "both samples were centrifuged"

Page 2, line 76: spacing in between "10minutes"

Page 2, line 79: "serum copper, zinc" can be replaced with "serum copper and zinc"

Page 2, line 76: please mention the term in which serum copper and zinc were measure. 

Page 2, line 85: please replace "students' t-test" with "student's t-test"

Page 3, line 91: delete "as"

Page 3, line 109, 110: please provide the reference for the statement

Page 3, line 115: delete "to be"

Page 4, line 142 and 146: add space between the text and reference

Page 5, line 177: abbreviate the name of the journal